# Food–Drug Interactions: Effect of Propolis on the Pharmacokinetics of Enrofloxacin and Its Active Metabolite Ciprofloxacin in Rabbits

**DOI:** 10.3390/ph18070967

**Published:** 2025-06-27

**Authors:** Ali Sorucu, Cengiz Gokbulut, Busra Aslan Akyol, Osman Bulut

**Affiliations:** 1Department of Pharmacology and Toxicology, Faculty of Milas Veterinary Medicine, Muğla Sitki Koçman University, 48200 Muğla, Türkiye; 2Beekeeping and Silkworm Research and Application Centre, Muğla Sıtkı Koçman University, 48200 Muğla, Türkiye; 3Department of Medical Pharmacology, Faculty of Medicine, Balikesir University, 10145 Balikesir, Türkiye; cgokbulut@gmail.com; 4Department of Veterinary Pharmacology and Toxicology, Institute of Health Sciences, Balikesir University, 10145 Balikesir, Türkiye; busraslan26@gmail.com; 5Department of Surgery, Faculty of Milas Veterinary Medicine, Mugla Sıtkı Koçman University, 48200 Muğla, Türkiye; obulut@mu.edu.tr

**Keywords:** propolis, drug interactions, pharmacokinetic, enrofloxacin, ciprofloxacin

## Abstract

Propolis is a natural resinous substance produced by honeybees that has many biological activities. For thousands of years, it has been widely used as a dietary supplement and traditional medicine to treat a variety of ailments due to its antimicrobial, anti-inflammatory, antioxidant, immunomodulatory, and wound-healing properties. Nutritional supplements and foods may interact with drugs both pharmacodynamically and pharmacokinetically, which could raise clinical concerns. **Background/Objectives**: This study aimed to investigate the effect of propolis on the plasma disposition of enrofloxacin and to assess the potential pharmacokinetic interaction in rabbits. **Methods**: In this study, enrofloxacin was applied per os (20 mg/kg) and IM (10 mg/kg) and with propolis (100 mg resin/kg) administration in four groups of rabbits (each of six individuals). Heparinized blood samples were collected at 0, 0.1, 0.3, 0.5, 1, 2, 4, 8, 12, and 24 h post-administration. HPLC-FL was used to analyze the plasma concentrations of enrofloxacin and its active metabolite ciprofloxacin following liquid–liquid phase extraction, i.e., protein precipitation with acetonitrile and partitioning with sodium sulfate. **Results**: The results revealed that propolis coadministration significantly affected the plasma disposition of enrofloxacin and its active metabolite after both per os and intramuscular administration routes. Significantly greater AUC (48.91 ± 11.53 vs. 26.11 ± 12.44 µg.h/mL), as well as longer T_1/2λz_ (11.75 ± 3.20 vs. 5.93 ± 2.51 h) and MRT (17.26 ± 4.55 vs. 8.96 ± 3.82 h) values of enrofloxacin and its metabolite ciprofloxacin, were observed after the coadministration of propolis compared to enrofloxacin alone following both per os and IM routes in rabbits. **Conclusions**: The concurrent use of propolis and prescription medications may prolong the half-life (T_1/2λz_) and increase the systemic availability of chronically used drugs with narrow therapeutic indices. The repeated use of drugs such as antibiotics, heart medications, and antidepressants, or drugs with a narrow therapeutic index such as antineoplastic and anticoagulant agents, can cause toxic effects by raising blood plasma levels. Considering the varied metabolism of rabbits and humans, further validation of this study may require thorough clinical trials in humans.

## 1. Introduction

A drug interaction occurs when two or more drugs, or a drug and a food or dietary supplement, affect each other’s actions. This interaction can reduce the systemic clearance of the drug or increase its bioavailability. As a result, there may be significant rises in the area under the plasma concentration–time curve (AUC) for the drug, leading to unexpected side effects [1]. The use of natural dietary supplements has surged worldwide at an unprecedented rate. Importantly, some components of natural extracts have been identified as substrates, inducers, and/or inhibitors of the transporters and/or enzymes responsible for the disposition of prescribed drugs [2]. Therefore, it is crucial to consider the potential risks associated with the combined use of dietary supplements and prescribed medications. This combination may compromise disease management and even increase the risk of adverse drug reactions.

Drug–food interactions frequently occur when drugs are taken with functional foods. These functional foods affect the treatment process by changing the effects of drugs or causing toxicity. The interaction most commonly occurs in the liver due to altered activity of the microsomal enzyme cytochrome P450 (CYP 450). When drugs are used together with the functional foods that induce their metabolism, their elimination rates increase, their terminal half-life (T_1/2λz_) shortens, and their therapeutic effects decrease [3]. If the drug used is a prodrug, it can rapidly turn into an active drug and cause toxic effects due to its sudden increase in plasma. When used in combination with foods that inhibit the drug metabolism, their T_1/2λz_ increases as their degradation decreases, and toxic and lethal effects may occur by exceeding the therapeutic window in cases of repeated doses. If the drug is a prodrug, it transforms into the main active drug form very late or not at all. As a result, the drug is prevented from reaching the therapeutic range, and the treatment cannot be achieved [3,4,5]. For example, natural products such as grapefruit, St. John’s wort, kava, spinach, black mulberry, broccoli, raspberry, pearl thorn, ginseng, and licorice root have been found to inhibit CYP 450 enzymes [5,6,7,8]. Foods such as valerian, echinacea, and garlic cause induction of CYP 450 enzymes [5,6,7,8]. Active substances, such as phenolic compounds, in these foods cause significant drug interactions by altering biotransformation enzyme activities. For example, grapefruit phenolic compounds such as furanocoumarins, naringin, naringenin, and hesperidin inhibit CYP 450 enzymes (particularly CYP 3A4) [5,6,7,8]. For this reason, many drugs metabolized by the CYP 3A4 enzyme contain a warning not to take them with grapefruit [5,6,7,8].

Propolis is a bee product formed by honeybees (Apis mellifera) by mixing substances such as plant resins collected from nature. More than 300 compounds, including phenolic compounds, have been identified in propolis. Propolis contains high amounts of phenolic compounds such as naringin, naringenin and hesperidin, which are responsible for grapefruit–drug interaction [9,10,11,12,13,14]. The activity of Brazilian green propolis on human CYP450 enzymes was investigated in vitro, and it was determined that propolis inhibited CYP1A2, CYP2C9, CYP2C19, and CYP3A4 enzymes [15]. It was found that kaempferol, which is particularly abundant in propolis, inhibited these enzymes [15]. The study determined in vitro that the phenolic compounds genistein, quercetin, and chrysin are found in high quantities in propolis, and they have a dose-dependent inductive effect on CYP1A1 activity. In contrast, quercetin has a dose-dependent inductive effect on CYP3A1 activity [16]. In another study, diosmetin and luteolin were found to inhibit the metabolism of midazolam by blocking CYP3A4 enzymes [17]. Quercetin, luteolin, kaempferol, naringenin, and CAPE, which are responsible for enzyme interaction, were found in large amounts in the analysis of propolis in Türkiye [14]. Akbay et al., 2017, investigated the interaction between propolis and warfarin and found that propolis significantly increased the international normalized ratio value, which is the pharmacodynamic activity of warfarin [18]. The study determined that propolis significantly altered the metabolism of duloxetine, a serotonin and norepinephrine reuptake inhibitor metabolized by CYP1A2 and CYP2D6 [19]. Some studies have suggested that the phenolic components in propolis may interact with drugs, which should be investigated in vivo and in vitro [20]. However, in this study, propolis and some drugs were given to 16 volunteers, and it was concluded that propolis did not change the pharmacokinetic parameters, although small changes were observed in the results obtained [21].

Enrofloxacin and ciprofloxacin are broad-spectrum fluoroquinolone group antibiotics. Ciprofloxacin is a metabolite of enrofloxacin, and ciprofloxacin is widely used in human health, while enrofloxacin is used in animal health. The antibiotics are metabolized by the CYP3ZA4 enzyme family [22].

This study aimed to investigate the effect of propolis on the pharmacokinetic dispositions of enrofloxacin following concurrent administration in rabbits. This research will provide valuable insights into the potential pharmacokinetic interactions between enrofloxacin and propolis, contributing to a deeper understanding of the safety and efficacy of the concurrent use of propolis with drugs.

## 2. Results

During this study, neither systemic nor local adverse effects were observed in any animal due to enrofloxacin or propolis administration. This study investigated whether there were any changes in pharmacokinetic parameters after the administration of propolis with oral (per os) and IM enrofloxacin treatment. In addition, the levels of ciprofloxacin, the metabolite of enrofloxacin, were measured, and the effect of propolis on biotransformation was investigated. The plasma samples collected in this study were analyzed using HPLC-FL for enrofloxacin and its metabolite ciprofloxacin. The mean and SD values of enrofloxacin and ciprofloxacin after IM enrofloxacin treatment are given in comparison with propolis treatment in Figure 1 and Appendix A. Similarly, the mean and SD values of enrofloxacin and ciprofloxacin after per os enrofloxacin treatment compared to propolis treatment are presented in Figure 1 and Appendix A. Significant differences were observed in the results obtained. The significance of the results was analyzed pharmacokinetically. T_1/2λz_: Terminal half-life, T_max_: Time to reach peak plasma concentration, C_max_: Peak plasma concentration, T_last_: Time to last detectable concentration, C_last_: Last detectable concentration, AUC_0–∞_: Area under the concentration–time curve from zero up to ∞ with extrapolation of the terminal phase, AUMC_0–∞_: Area under the first moment of the concentration–time curve from zero up to ∞ with extrapolation of the terminal phase, and MRT_0–∞_: Mean residence time from zero up to ∞ with extrapolation of the terminal phase differences are shown statistically (Table 1, Table 2, Table 3 and Table 4).

### 2.1. Evaluation of Enrofloxacin and Ciprofloxacin Results After IM Enrofloxacin and Per os Propolis

The effect of propolis on enrofloxacin in plasma after IM treatment with enrofloxacin is shown in Table 1. In the results obtained, propolis did not cause any significant change, except for the T_1/2λz_ and MRT_0–∞_ of enrofloxacin. Propolis administration increased T_1/2λz_ and MRT_0–∞_ (h) values were twofold. Although the bioavailability of enrofloxacin increased almost twofold, it was not statistically significant.

**Table 1 pharmaceuticals-18-00967-t001:** Mean (±SD) pharmacokinetic parameters of enrofloxacin following intramuscular administration of enrofloxacin (ENR, 10 mg/kg) alone or with propolis (PP) administered per os at 100 mg resin/kg in rabbits (*n* = 6).

Intramuscular		
Parameters	ENR	ENR + PP
T_1/2λz_ (h)	2.88 ± 0.59	5.64 ± 1.63 *
T_max_ (h)	0.47 ± 0.08	0.52 ± 0.29
C_max_ (µg/mL)	2.98 ± 0.35	2.90 ± 0.45
T_last_ (h)	24.00 ± 0.00	24.00 ± 0.00
C_last_ (µg/mL)	0.04 ± 0.01	0.17 ± 0.15
AUC_0–∞_ (µg.h/mL)	12.59 ± 2.34	21.18 ± 8.86
AUMC_0–∞_ (µg.h^2^/mL)	59.21 ± 18.78	180.15 ± 143.37
MRT_0–∞_ (h)	4.63 ± 0.69	7.72 ± 2.53 *

ENR: Enrofloxacin, PP: Propolis, T_1/2λz_: Terminal half-life, T_max_: Time to reach peak plasma concentration, C_max_: Peak plasma concentration, T_last_: Time to last detectable concentration, C_last_: Last detectable concentration, AUC_0–∞_: Area under the concentration–time curve from zero up to ∞ with extrapolation of the terminal phase, AUMC_0–∞_: Area under the first moment of the concentration–time curve from zero up to ∞ with extrapolation of the terminal phase, MRT_0–∞_: Mean residence time from zero up to ∞ with extrapolation of the terminal phase. * *p* < 0.05 (statistically different from enrofloxacin).

The effect of propolis on the metabolism of enrofloxacin to ciprofloxacin after IM treatment with enrofloxacin and the pharmacokinetic parameters of ciprofloxacin are shown in Table 2. Propolis treatment caused statistically significant differences, apart from in the T_max_ and C_max_ values. It was observed that propolis administration increased the T_1/2λz_ and mean residence time (MRT) of ciprofloxacin by a factor of two, similar to enrofloxacin. There was also a significant increase in the AUC_0_. Considering the AUC value of ciprofloxacin, it seems that the administration of propolis increased the biotransformation of enrofloxacin. However, when the AUC values of enrofloxacin are compared, it is clearly seen that the biotransformation decreased (Table 2). While circulating (AUC) enrofloxacin 12.59 µg was metabolized to 1.88 µg ciprofloxacin (14.9%), circulating (AUC) enrofloxacin + propolis 21.18 µg was metabolized to 2.71 µg ciprofloxacin (12.79%) (Table 2 and Table 3).

**Table 2 pharmaceuticals-18-00967-t002:** Mean (±SD) pharmacokinetic parameters of ciprofloxacin following intramuscular administration of enrofloxacin (ENR, 10 mg/kg) alone or with propolis (PP) administered per os at 100 mg resin/kg in rabbits (*n* = 6).

Intramuscular		
Parameters	CPR	CPR + PP
T_1/2λz_ (h)	4.97 ± 1.34	8.74 ± 1.93 **
T_max_ (h)	2.33 ± 0.82	2.40 ± 0.89
C_max_ (µg/mL)	0.21 ± 0.03	0.20 ± 0.06
T_last_ (h)	24.00 ± 0.00	24.00 ± 0.00
C_last_ (µg/mL)	0.01 ± 0.01	0.03 ± 0.01 **
AUC_0–∞_ (µg.h/mL)	1.88 ± 0.31	2.71 ± 0.45 *
AUMC_0–∞_ (µg.h^2^/mL)	14.87 ± 5.22	34.97 ± 6.60 **
MRT_0–∞_ (h)	7.75 ± 1.45	13.08 ± 2.74 **

CPR: Ciprofloxacin, PP: Propolis, T_1/2λz_: Terminal half-life, T_max_: Time to reach peak plasma concentration, C_max_: Peak plasma concentration, T_last_: Time to last detectable concentration, C_last_: Last detectable concentration, AUC_0–∞_: Area under the concentration–time curve from zero up to ∞ with extrapolation of the terminal phase, AUMC_0–∞_: Area under the first moment of the concentration–time curve from zero up to ∞ with extrapolation of the terminal phase, MRT_0–∞_: Mean residence time from zero up to ∞ with extrapolation of the terminal phase. ** *p* < 0.01; * *p* < 0.05 (statistically different from ciprofloxacin).

### 2.2. Evaluation of Enrofloxacin and Ciprofloxacin Results After Per os Enrofloxacin and Propolis

The effect of propolis on the level of enrofloxacin in the blood after oral administration of enrofloxacin is shown in Table 3. Propolis was found to almost double the T_1/2λz_ of enrofloxacin, increasing it from 5.93 to 11.75. The AUC_0–∞_ value was increased by 53%, and propolis significantly increased the systemic availability of enrofloxacin. In addition, the MRT_0–∞_ results showed that propolis prolonged considerably (51%) the presence of enrofloxacin in the blood. The results show that propolis prolonged the T_1/2λz_ of per os enrofloxacin and increased considerably its systemic availability.

**Table 3 pharmaceuticals-18-00967-t003:** Mean (±SD) pharmacokinetic parameters of enrofloxacin following per os administration of enrofloxacin (ENR, 20 mg/kg) alone or with propolis (PP) administered per os at 100 mg resin/kg in rabbits *(n* = 6).

Per os		
Parameters	ENR	ENR + PP
T_1/2λz_ (h)	5.93 ± 2.51	11.75 ± 3.20 **
T_max_ (h)	2.00 ± 1.10	3.08 ± 2.65
C_max_ (µg/mL)	2.38 ± 0.34	2.65 ± 0.65
T_last_ (h)	24.00 ± 0.00	24.00 ± 0.00
C_last_ (µg/mL)	0.23 ± 0.24	0.70 ± 0.21 **
AUC_0–∞_ (µg.h/mL)	26.11 ± 12.44	48.91 ± 11.53 **
AUMC_0–∞_ (µg.h^2^/mL)	272.94 ± 237.10	876.29 ± 399.59 *
MRT_0–∞_ (h)	8.96 ± 3.82	17.26 ± 4.55 **

ENR: Enrofloxacin, PP: Propolis, T_1/2λz_: Terminal half-life, T_max_: Time to reach peak plasma concentration, C_max_: Peak plasma concentration, T_last_: Time to last detectable concentration, C_last_: Last detectable concentration, AUC_0–∞_: Area under the concentration–time curve from zero up to ∞ with extrapolation of the terminal phase, AUMC_0–∞_: Area under the first moment of the concentration–time curve from zero up to ∞ with extrapolation of the terminal phase, MRT_0–∞_: Mean residence time from zero up to ∞ with extrapolation of the terminal phase. ** *p* < 0.01; * *p* < 0.05 (statistically different from enrofloxacin).

The effect of propolis on the biotransformation of enrofloxacin to ciprofloxacin after per os enrofloxacin treatment and its pharmacokinetic parameters are given in Table 4. Propolis treatment significantly changed the effect on the T_1/2λz_, C_max_, AUMC_0–∞_, and MRT_0–∞_ parameters of ciprofloxacin. Propolis significantly prolonged the T_1/2λz_ (44%) and MRT_0–∞_ (45%) of ciprofloxacin. In addition, the C_max_ level was reduced considerably with propolis administration. This suggests that enrofloxacin is metabolized more slowly to ciprofloxacin. While circulating (AUC) enrofloxacin 26.11 µg was metabolized to 4.68 µg ciprofloxacin (18%), circulating (AUC) enrofloxacin + propolis 48.91 µg was metabolized to 5.23 µg ciprofloxacin (10%) (Table 3 and Table 4). The decreased AUC ratio and C_max_ indicate that propolis reduces the biotransformation of enrofloxacin to ciprofloxacin.

**Table 4 pharmaceuticals-18-00967-t004:** T Mean (±SD) pharmacokinetic parameters of ciprofloxacin following per os administration of enrofloxacin (ENR, 20 mg/kg) alone or with propolis (PP) administered per os at 100 mg resin/kg in rabbits (*n* = 6).

Per os		
Parameters	CPR	CPR + PP
T_1/2λz_ (h)	7.22 ± 2.34	16.35 ± 4.72 **
T_max_ (h)	4.76 ± 1.63	5.33 ± 2.07
C_max_ (µg/mL)	0.36 ± 0.09	0.20 ± 0.08 *
T_last_ (h)	24.00 ± 00.00	24.00 ± 0.00
C_last_ (µg/mL)	0.05 ± 0.03	0.08 ± 0.02
AUC_0–∞_ (µg.h/mL)	4.68 ± 0.98	5.23 ± 1.41
AUMC_0–∞_ (µg.h^2^/mL)	55.45 ± 27.43	127.97 ± 34.55 **
MRT_0–∞_ (h)	11.39 ± 3.82	24.99 ± 6.51 **

CPR: Ciprofloxacin, PP: Propolis, T_1/2λz_: Terminal half-life, T_max_: Time to reach peak plasma concentration, C_max_: Peak plasma concentration, T_last_: Time to last detectable concentration, C_last_: Last detectable concentration, AUC_0–∞_: Area under the concentration–time curve from zero up to ∞ with extrapolation of the terminal phase, AUMC_0–∞_: Area under the first moment of the concentration–time curve from zero up to ∞ with extrapolation of the terminal phase, MRT_0–∞_: Mean residence time from zero up to ∞ with extrapolation of the terminal phase. ** *p* < 0.01; * *p* < 0.05 (statistically different from ciprofloxacin).

The dose-optimized pharmacokinetic parameters comparing enrofloxacin and ciprofloxacin are given in Appendix A. In the statistical analysis performed with dose optimization, the AUC values of IM enrofloxacin with per os enrofloxacin and IM enrofloxacin + propolis with per os enrofloxacin + propolis did not differ significantly, whereas a significant difference was observed in the ciprofloxacin comparisons (Appendix A).

## 3. Discussion

Although enrofloxacin was administered at a higher dose via the per os route (20 mg/kg) compared to the intramuscular route (10 mg/kg), the terminal half-life (T_1/2λz_) was significantly longer after per os administration than after intramuscular administration. Additionally, while the C_max_ and AUC values were not significantly different (2.38–2.98 µg/mL for C_max_ and 13.05–12.59 µg.h/mL for the AUC after dose normalization), the T_max_ (2.00–0.47 h) and MRT (8.96–4.63 h) values of enrofloxacin alone were significantly longer for the per os route compared to the intramuscular route. Similar results were also obtained after concurrent administrations of propolis and enrofloxacin following per os and intramuscular routes. These results suggested that absorption is the rate-limiting process in the kinetics of per os administration, and the terminal half-life represents the half-life of absorption of enrofloxacin in rabbits. This phenomenon is termed the flip-flop effect, in which the disposition of the drug from the body is controlled by the absorption process [23]. Furthermore, the similarity of the AUC values after oral (13.05 µg.h/mL) and intramuscular (12.59 µg.h/mL) administration of enrofloxacin, following dose normalization, may suggest that the oral bioavailability of enrofloxacin is comparable to that of the intramuscular route.

The results obtained in the present study indicated that the AUC, T_1/2λz_, and MRT values obtained after concurrent administration of propolis with enrofloxacin were found to be significantly higher and longer when compared with those obtained after the administration of enrofloxacin alone, respectively. Furthermore, the concurrent administration of propolis significantly affected the plasma kinetic disposition of ciprofloxacin, the active metabolite of enrofloxacin. Despite the C_max_ (2.90–2.98 µg/mL) and T_max_ (0.52–0.47 h) values following intramuscular administration of enrofloxacin with and without propolis being as they are, the half-life (T_1/2λz_), mean residence time (MRT), and area under the curve (AUC) values were significantly longer when enrofloxacin was administered in combination with propolis compared to enrofloxacin alone. Although the exact mechanism of action of propolis remains unclear, these results suggest that concurrent propolis administration significantly reduces elimination and increases the bioavailability of enrofloxacin following both per os and intramuscular routes.

The AUC values indicate that propolis increases the systemic availability of both enrofloxacin and ciprofloxacin. Although the AUC seems to be high when propolis is administered, it is seen that the biotransformation of enrofloxacin to ciprofloxacin is proportionally reduced. This indicates that propolis inhibits the CYP450 enzymes that metabolize enrofloxacin. The interaction appears to be more significant with the per os administration of enrofloxacin.

There are many studies on the pharmacodynamic interaction of propolis, but very few on the pharmacokinetic aspect [24]. In a study by Cusinato et al., subtherapeutic doses of caffeine, losartan, omeprazole, metoprolol, midazolam, and fexofenadine were administered to 16 volunteers with propolis (375 mg/day) and evaluated for bioavailability by considering the AUC. The fexofenadine, caffeine, and losartan AUC decreased, whereas the omeprazole and midazolam AUC increased. Since the detected increases and decreases were less than 20%, it was concluded that propolis has no drug interaction potential. T_1/2λz_ was not evaluated in this study [21]. Here, in contrast to this study, 100 mg of propolis administered to rats increased the T_1/2λz_ and AUC values of enrofloxacin and ciprofloxacin almost twofold. It was concluded that propolis can cause serious drug effects. In another study, propolis was coadministered with duloxetine, and the pharmacokinetic parameters were studied. A 20–30% increase in C_max_ and AUC values was observed [19]. The study concluded that propolis may cause drug interactions similar to those observed in the present study. No assessment of biotransformation was made, as the metabolites of duloxetine were not analyzed in this study [19]. Although there are insufficient studies on the pharmacokinetic efficacy of propolis, it has been predicted that propolis interacts with the CYP3A enzyme family, which is responsible for the metabolism of drugs. Therefore, there is potential interaction with all drugs biotransformed by this family of enzymes [15,16,17]. The results of the present study are proof of the predictions of these studies. In addition, many pharmacodynamic studies have revealed that propolis increases the efficacy of drugs, which may be due to an increase in the drug concentration in the blood and the systemic availability of the drug [25,26,27]. It has also been predicted that some drugs with a narrow therapeutic range, such as warfarin, may interact pharmacodynamically with some drugs and may cause toxic effects [18,24]. This may be due to prolonging the T_1/2λz_ and MTR of the drug, increasing toxic levels with repeated doses, or increasing blood levels to toxic levels due to the increased systemic availability (AUC) of the drug. The results of the present study indicate that these possibilities may occur.

The potential drug interaction associated with propolis is primarily due to its phenolic compounds. These compounds, which include furanocoumarins, naringin, naringenin, and hesperidin, have been reported to interact with various drugs [5,6,7,8]. Specifically, they tend to prolong these drugs’ half-life (T_1/2λz_) by inhibiting certain CYP450 enzymes [5,6,7,8]. The current study observed a significant increase in the drugs’ T_1/2λz_ and systemic availability. This increase is likely attributable to the phenolic compounds present in propolis. Although these compounds could be identified and analyzed to determine their effects, it is important to note that propolis is a complex substance containing over 300 different compounds. As a result, pinpointing the specific substance responsible for the interaction is quite challenging.

## 4. Materials and Methods

### 4.1. Reagents, Solvents, and Materials

The pure analytical standard of enrofloxacin (purity: 99.9%, CAS Number: 93106-60-6), ciprofloxacin (purity: ≥98%, CAS Number: 85721-33-1), and marbofloxacin (purity: ≥98, CAS Number: 115550-35-1) were obtained from Sigma-Aldrich (Chemie, Steinheim, Germany). HPLC grade acetonitrile Sigma-Aldrich 34851 (Chemie, Steinheim, Germany), potassium phosphate monobasic (purity: ≥99.0%, CAS Number: 7778-77-0), sodium sulfate anhydrous (purity: ≥99.0%, CAS Number: 7757-82-6), and trifluoroacetic acid (purity: ≥90–100%, CAS Number: 76-05-1) were supplied by Sigma-Aldrich (Chemie, Steinheim, Germany). Ultrapure water (18.2 MΩ.cm at 25 °C) for the preparation of the mobile phase and analytical procedures was produced using the Millipore Simplicity^®^ water purification system (Billerica, MA, USA).

### 4.2. Propolis Collection and Extraction

Propolis traps were placed in the hives of an apiary in the Marmara region of Türkiye, and about 1000 g of reddish-colored propolis was collected in May (Appendix A). The collected propolis was kept in a deep freezer, homogenized to powder using a grinder, and stored at −20 °C until the extraction stage. A total of 600 g of homogenized propolis was weighed and dissolved in 1800 mL of hydro-alcoholic solvent (ethyl alcohol 70% and water 30%) for propolis extraction [14]. The propolis was kept in the solvent in an orbital shaker for one hour and then in an ultrasonic bath for half an hour to dissolve it thoroughly. The propolis extract was first filtered through standard filter paper, then kept at +4 °C for 1 h and prepared by filtering through Whatman No. 1 filter paper again. To determine the resin ratio of this propolis extract, 2 mL of liquid extract was taken into a tared glass tube, and the solvent was evaporated with the help of a vacuum centrifuge (Jouan, RC 10-10). The remaining resin was weighed, and the main stock extract was determined to be 1 mL of 390 mg resin. The prepared propolis solution was diluted with solvent, and the stock was prepared as 200 mg resin/mL, transferred to glass bottles, and stored at +4 °C until the pharmacokinetic study.

### 4.3. Pharmacokinetic Experimental Design

This study was conducted at Muğla Sıtkı Koçman University Experimental Animals Application and Research Centre in Muğla in Türkiye, after obtaining permission from the local Ethics Committee for Animal Experiments of Muğla Sıtkı Koçman University, under the approval number 16/20 (approved on 9 November 2020). This study was carried out under ideal conditions for rabbits, with ad libitum food and water. A total of 12 New Zealand White female rabbits, aged 4–6 months and weighing 2500–3000 g, were used in this study. The rabbits were randomly allocated into two groups, and each group consisted of six animals in the first phase of the study. This study was performed according to a two-phase longitudinal pharmacokinetic design. A 40-day elimination period was allowed between the treatments. Per os enrofloxacin was carried out in the first phase, and intramuscular (IM) enrofloxacin was carried out in the second phase of the study.

Propolis was administered by per os gavage to the rabbits of the propolis groups in this study as follows: the 1st-day treatment dose ¼, the 2nd-day treatment dose ½, and the 3rd and 4th-day treatment dose (100 mg resin/kg).

The rabbits treated with enrofloxacin only (IM-per-os) also received 0.5 mL/kg of propolis extraction solvent (PES), which was ethanol 70%/water 30% for a control.

**Group 1:** Per os enrofloxacin 20 mg/kg + PES.**Group 2:** Per os enrofloxacin 20 mg/kg + per os propolis 100 mg resin/kg.**Group 3:** Intramuscular (IM) enrofloxacin 10 mg/kg + PES.**Group 4:** IM enrofloxacin 10 mg/kg + per os propolis 100 mg resin/kg.

Blood samples were collected from the *Vena auricularis* of each rabbit at 0, 0.1, 0.3, 0.5, 1, 2, 4, 8, 12, and 24 h after enrofloxacin and enrofloxacin + propolis administrations into heparinized tubes. The collected blood samples were centrifuged at 13,000 rpm for 10 min, and plasma was removed and stored at −20 °C until HPLC analysis.

### 4.4. Analytical Procedure

#### 4.4.1. Instrumentation

Chromatographic analyses were performed using an Agilent 1260 HPLC system (Agilent, Waldbronn, Germany) consisting of a binary high-pressure gradient system used to analyze enrofloxacin and its metabolite ciprofloxacin. An Agilent binary pump (G1312B) was used to deliver the mobile phase to the analytical column. Sample injection was performed via an Agilent autosampler (G1367E) coupled with an injection valve (Rheodyne^®^, St. Louis, MO, USA) equipped with a 100 µL variable loop. Detection was achieved with a fluorometric detector (G1321B), in compliance with data acquisition ChemStation^®^ Software C.01.08 by Agilent (Waldbronn, Germany). An Agilent vacuum degasser unit (G4225A) achieved degassing of the mobile phase. Operations and functions of the whole HPLC system were controlled with ChemStation^®^ Software (Agilent, Waldbronn, Germany).

All evaporations following the extraction of the samples were performed at 50 °C with a sample concentrator (Maxi-dry^®^ plus, Heto Lab. Equipment, Allerød, Denmark). A vortex (622, Isolab, Wertheim, Germany), a microcentrifuge (Mikro 200, Hettich, Zentrifugen, Tuttlingen, Germany), and an ultrasonic bath, Elmasonic^®^ S40-H (Elma, Singen, Germany), were employed for sample pre-treatment and extraction procedures.

#### 4.4.2. Chromatographic Conditions

The chromatographic conditions of HPLC for the analysis of enrofloxacin and its metabolite ciprofloxacin were used as described by Sekkin et al. (2012) [28]. An analytical column (Zorbax, Eclipse Plus^®^ C18, 5 µm, 250 mm × 4.6 mm, Agilent, Waldbronn, Germany) with a Nucleosil C18 guard column (Phenomenex, UK)—kept at 50 °C during the analysis in a column oven (G1316A)—was used for the separation of enrofloxacin and ciprofloxacin. The mobile phase consisted of acetonitrile, and methanol (50:50) and potassium phosphate monobasic (20 mM, pH: 3.40) were delivered in an isocratic fashion at a flow rate of 1 mL/min. The fluorescence detector (FL, G1321B, Agilent, Germany) was at an excitation wavelength of 280 nm and an emission wavelength of 450 nm. The Photo Diode Array Detector (DAD) was set at a wavelength of 279 nm for marbofloxacin (I.S.) analysis. The injection volume was 50 µL throughout the analysis, and a chromatographic analysis was completed in 10 min for plasma samples following injection.

#### 4.4.3. Preparation of Standard Solutions

Stock analytical standard solutions (50 µg/mL) of enrofloxacin and ciprofloxacin and the internal standards (I.S.) marbofloxacin (50 µg/mL) were prepared in acetonitrile and stored in glass bottles at 4 °C. The calibration and working standard solutions were prepared in acetonitrile, a dilution of the stock standard solutions. These were diluted with acetonitrile to give solutions of 0.1, 0.5, 1, 5, and 10 µg/mL. These solutions were used to spike drug-free plasma at different levels to generate standard curves and determine the recoveries of the extraction procedure.

#### 4.4.4. Sample Preparations and Extraction Procedures

The plasma concentrations of enrofloxacin and ciprofloxacin were determined by HPLC-FL in combination following liquid–liquid phase extraction procedures, according to Anastassiades et al. (2003) and Sekkin et al. (2012), with minor modifications, as described below [28,29]. Accordingly, blank plasma samples (0.2 mL) were spiked with 20–40 µL of enrofloxacin and ciprofloxacin standard solution to reach the following final concentrations: 0, 0.01, 0.05, 0.1, 0.5, 1, 5, and 10 µg/mL. The plasma samples (spiked and experimental) were combined with 25 µL of the internal standard (marbofloxacin, 5 µg/mL). Subsequently, 0.8 mL of acetonitrile was added for the deproteinization of the plasma samples, which were vortexed for 1 min and centrifuged at 12,000 rpm for 5 min. The supernatant was transferred to another 2 mL plastic tube. Sodium sulfate anhydrous (0.05 g) was added, vortexed for 1 min, and then centrifuged at 12,000 rpm for 5 min. The upper organic phase was transferred to a 10 mL glass tube and evaporated in a vacuum concentrator (Maxi-dry^®^ plus, Heto Lab. Equipment, Allerød, Denmark) at 50 °C. The completely dried residue was dissolved with 200 µL mobile phase and vortexed for 15 s. Finally, this solution (50 μL) was injected into the HPLC-FL system for analysis. The chromatograms of enrofloxacin and ciprofloxacin are given in Appendix A.

### 4.5. Validation of Analytical Method

Analytical validation was used to determine enrofloxacin and ciprofloxacin in plasma samples according to the International Conference on Harmonization (ICH) guidelines for the validation of analytical procedures [30].

The analyte was identified with the retention times of the pure reference standard (Appendix A). Recoveries of the molecules under study were measured by the comparison of the peak areas from spiked plasma samples with the areas resulting from direct injections of standards prepared in acetonitrile. The inter- and intra-assay precisions of the extraction and chromatography procedures were evaluated by processing replicate aliquots of drug-free rabbit plasma samples containing known amounts of the drugs on different days. The calibration graphs for enrofloxacin and ciprofloxacin were prepared (linear range 0.1–50 µg/mL for plasma analysis). The slope of the lines between the peak areas and drug concentration was determined by least squares linear regression, and the correlation coefficient (r) and coefficient of variations (CV) were calculated. Linearity was established to determine the relationship between enrofloxacin and ciprofloxacin concentration/detector response. The detection limits of enrofloxacin and ciprofloxacin were established with the HPLC-FL analysis of blank plasma samples fortified with the standard, measuring the baseline noise at the retention time of the peak. The mean baseline noise at the peak retention time plus three standard deviations (SDs) was defined as the detection limit (LOD). The mean baseline noise plus six SDs was defined as the limit of quantification (LOQ).

The validation parameters of enrofloxacin and ciprofloxacin for the analysis of plasma samples are given in Table 5. The pharmacokinetic analysis did not consider plasma concentration values lower than the LOQ values.

### 4.6. Pharmacokinetics and Statistical Analysis of Data

The plasma concentration–time curves obtained after each treatment in the individual animals were fitted with the WinNonlin software program (Version 5.2, Pharsight Corporation, Mountain View, CA, USA). The pharmacokinetic parameters for each animal were analyzed using non-compartmental model analysis. The maximum plasma concentration (C_max_) and time to reach the maximum concentration (T_max_) were obtained from the plotted concentration–time curve of the drug in each animal. The trapezoidal rule was used to calculate the area under the plasma concentration–time curve (AUC).

The mean residence time (MRT) was calculated as follows: MRT_0→∞_ = AUMC_0→∞_/AUC_0→∞_

Terminal half-life (T_1/2λz_) was calculated as follows: T_1/2λz_= −ln(2)/λz
where λz represents the first-order rate constant associated with the terminal (log-linear) portion of the curve.

The pharmacokinetic parameters are reported as mean (±SD). Harmonic means were calculated for T_1/2λz_ and MRT_0–∞_. The normality of the data was tested using the Shapiro–Wilk test. Non-normally distributed data were analyzed by a non-parametric Mann–Whitney U-Test; meanwhile, normally distributed data were compared by independent *t*-test using SPSS Statistics 23.0 (IBM Corp, Armonk, NY, USA). A value of *p* < 0.05 and *p* < 0.01 was considered statistically significant.

## Figures and Tables

**Figure 1 pharmaceuticals-18-00967-f001:**
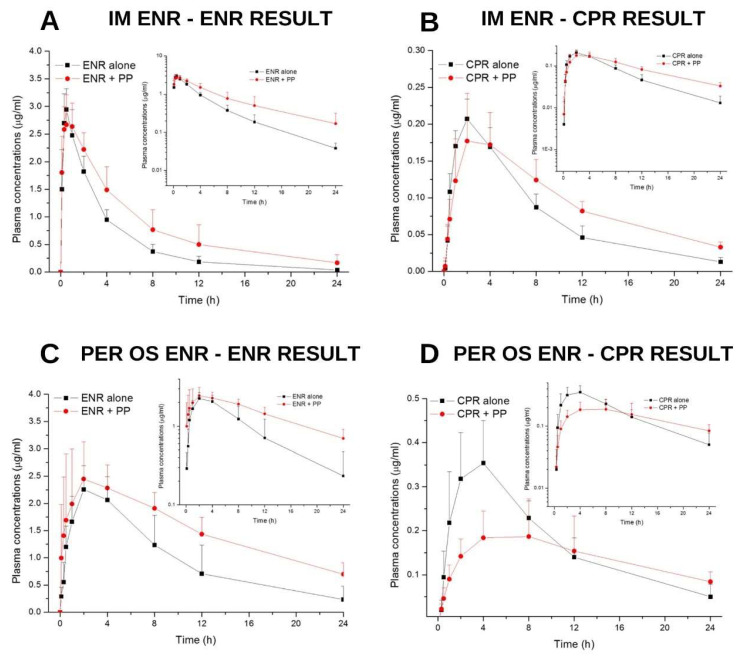
Time-dependent mean plasma plots of enrofloxacin and ciprofloxacin. (**A**): Time-dependent plasma ENR (enrofloxacin) graph (µg) after IM ENR treatment and IM ENR+ per os propolis administration. (**B**): Time-dependent plasma CPR (ciprofloxacin) graph (µg) after IM ENR treatment and IM ENR+ per os propolis administration. (**C**): Plasma ENR graph after per os ENR treatment and per os ENR + propolis administration. (**D**): Plasma CPR graph after per os ENR treatment and per os ENR+ propolis administration.

**Table 5 pharmaceuticals-18-00967-t005:** Validation parameters of the analytical method used to determine enrofloxacin (ENR) and ciprofloxacin (CPR) in plasma samples.

Parameters	ENR	CPR
**LOD (µg/mL)**	0.0032	0.0030
**LOQ (µg/mL)**	0.01	0.01
**Range of linearity (µg/mL)**	0.01–10	0.01–10
**Linearity (R^2^)**	0.9977–0.9995	0.9992–0.9998
**Recovery (%)**	98.08 (1.05)	82.88 (5.03)
**Coefficient of variation (%)**	3.44 (0.73)	4.74 (0.59)
**Inter-day precision (RSD%)**	1.02–1.34	1.10–1.22
**Intra-day precision (RSD%)**	0.56–0.98	0.23–0.75

ENR: Enrofloxacin; CPR: Ciprofloxacin; LOD: Limit of detection; LOQ: Limit of quantification; R^2^: Correlation coefficient. Values in the brackets represent the standard deviations for the recovery assays (n = 8).

## Data Availability

The original contributions presented in this study are included in the article/Appendix A. Further inquiries can be directed to the corresponding author(s).

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
