# Peer review of "Food–Drug Interactions: Effect of Propolis on the Pharmacokinetics of Enrofloxacin and Its Active Metabolite Ciprofloxacin in Rabbits"

_pharmaceuticals, 2025, doi:10.3390/ph18070967_

Round 1
Reviewer 1 Report
Comments and Suggestions for Authors
The study presents an important investigation into the pharmacokinetic interaction between propolis and enrofloxacin in rabbits. Novelty of the article is much appreciated. Writing is very clear. This is a promising study with significant clinical implications for drug safety. It could be a strong contribution to the pharmacokinetics field. However, many revisions are required. Correct validation protocol should be followed
- In line 19, in the abstract, please write specific diseases [to treat a variety of ailments ] such as antimicrobial, anti-inflammatory, antioxidant, immunomodulatory, and wound-healing properties.
- Write full names for all abbreviations when first mentioning such as os, AUC, MRT
- In the abstract, mention sampling time points more clearly (e.g., "Blood samples were collected at 1, 2, 4, 6, 8, 12, and 24h post-administration").
- In the abstract, write about extraction procedures of drugs from plasma [simply and precisely ] . protein precipitations or liquid extract procedures.
- In the abstract results subsection, add more numerical values to pharmacokinetic data with and without Propolis.
- In the abstract, conclusion subsection, mention specific toxic and serious adverse complications.
- In line 86, write full name for INR
- In the introduction, please refer to the old reported HPLC-FL for enrofloxacin such as
Araneda, C., Villar, P., Cuadros, C., Valle, M., Nunes, P. and Santelices, M., 2013. Single and multiple pharmacokinetics of enrofloxacin and ciprofloxacin in pigs. J. Bioequiv. Availab, 5(1), pp.041-046.
Cornejo, J., Lapierre, L., Iragüen, D., Cornejo, S., Cassus, G., Richter, P. and San Martin, B., 2012. Study of enrofloxacin and flumequine residues depletion in eggs of laying hens after oral administration. Journal of Veterinary Pharmacology and Therapeutics, 35(1), pp.67-72.
Are there any novelty in HPLC-FL method compared to old ones?
- In line 296, write country name after in the Marmara region. Also, at line 314 for the university. Also, for funding statement
- In line 390, the authors should follow validation protocol for bioanalysis (plasma, serum, etc.), FDA (2018), EMA (2011), or ICH M10 (2022) bioanalytical guidelines apply. https://www.ema.europa.eu/en/ich-m10-bioanalytical-method-validation-scientific-guideline
- Regarding 4.4.2. Chromatographic conditions, is it new method or old reported one , add reference if it is old procedure
- Provide reasons for choosing marbofloxacin as internal standard (I.S.) and why it was not measured using fluorescence detector ?
- In line 396, demonstrate results of The inter- and intra-assay precisions in new table
- In table 5, correct Linearity (r 2 ) to be R² (coefficient of determination)
- Add missed conclusion
- Add future plan and study limitations (e.g., " Results in rabbits may not directly translate to humans").
Author Response
Author responses to the editor and reviewer’ comments
Thanks to the Editor and the reviewers for these precious comments and suggestions. These comments are all valuable and very helpful for revising and improving our paper. We have studied all the comments carefully and have made corrections which we hope to meet with approval. Revised portions are highlighted for the 1. Refree as yellow, for the 2. as green, for the editör as red in the manuscript
Kindest regards
The authors of the manuscript
Referee 1
The study presents an important investigation into the pharmacokinetic interaction between propolis and enrofloxacin in rabbits. Novelty of the article is much appreciated. Writing is very clear. This is a promising study with significant clinical implications for drug safety. It could be a strong contribution to the pharmacokinetics field. However, many revisions are required. Correct validation protocol should be followed.
- In line 19, in the abstract, please write specific diseases [to treat a variety of ailments ] such as antimicrobial, anti-inflammatory, antioxidant, immunomodulatory, and wound-healing properties.
Reply: The regulation was made in accordance with the referee's recommendation.
- Write full names for all abbreviations when first mentioning such as os, AUC, MRT
Reply: The last section of the article contains a section titled ‘Abbreviations,’ which explains all abbreviations.
In accordance with the referee's suggestion, T1/2λz was added, and the abbreviations were added to the first places they appeared, also added to the per os abbreviations section.
- In the abstract, mention sampling time points more clearly (e.g., "Blood samples were collected at 1, 2, 4, 6, 8, 12, and 24h post-administration").
Reply: Blood collection times added to abstract
- In the abstract, write about extraction procedures of drugs from plasma [simply and precisely ] . protein precipitations or liquid extract procedures.
Reply: The following statement was added to the abstract.
“HPLC-FL analysed the plasma concentrations of enrofloxacin and its active metabolite ciprofloxacin following liquid-liquid phase extraction, i.e. protein precipitation with acetonitrile and partitioning with sodium sulfate”
- In the abstract results subsection, add more numerical values to pharmacokinetic data with and without Propolis.
Reply: More numerical values to pharmacokinetic data with and without Propolis were added to the abstract.
- In the abstract, conclusion subsection, mention specific toxic and serious adverse complications.
Reply: The regulation was made in accordance with the referee's recommendation.
- In line 86, write full name for INR
Reply: The regulation was made in accordance with the referee's recommendation.
- In the introduction, please refer to the old reported HPLC-FL for enrofloxacin such as
Araneda, C., Villar, P., Cuadros, C., Valle, M., Nunes, P. and Santelices, M., 2013. Single and multiple pharmacokinetics of enrofloxacin and ciprofloxacin in pigs. J. Bioequiv. Availab, 5(1), pp.041-046.
Cornejo, J., Lapierre, L., Iragüen, D., Cornejo, S., Cassus, G., Richter, P. and San Martin, B., 2012. Study of enrofloxacin and flumequine residues depletion in eggs of laying hens after oral administration. Journal of Veterinary Pharmacology and Therapeutics, 35(1), pp.67-72.
Are there any novelty in HPLC-FL method compared to old ones?
Reply: The HPLC method of the present study was similar, but the extraction method was different.
Reply: Since this study does not focus on method development and validation, the following statement is added to the "materials and methods" section instead of the "introduction".
"As enrofloxacin and its metabolite ciprofloxacin are compounds with fluorescent properties, HPLC-FLD technique is widely employed to determine the levels of these compounds with greater sensitivity and specificity in pharmacokinetic studies and residue analyses (Araneda et al., 2013, Cornejo et al., 2012)".
Araneda, C., Villar, P., Cuadros, C., Valle, M., Nunes, P. and Santelices, M., 2013. Single and multiple pharmacokinetics of enrofloxacin and ciprofloxacin in pigs. J. Bioequiv. Availab, 5(1), pp.041-046.
Cornejo, J., Lapierre, L., Iragüen, D., Cornejo, S., Cassus, G., Richter, P. and San Martin, B., 2012. Study of enrofloxacin and flumequine residues depletion in eggs of laying hens after oral administration. Journal of Veterinary Pharmacology and Therapeutics, 35(1), pp.67-72.
- In line 296, write country name after in the Marmara region. Also, at line 314 for the university. Also, for funding statement
Reply: The regulation was made in accordance with the referee's recommendation.
- In line 390, the authors should follow validation protocol for bioanalysis (plasma, serum, etc.), FDA (2018), EMA (2011), or ICH M10 (2022) bioanalytical guidelines apply. https://www.ema.europa.eu/en/ich-m10-bioanalytical-method-validation-scientific-guideline
Reply: The protocol suggested by the referee for validation was cited in the relevant place in the article.
- Regarding 4.4.2. Chromatographic conditions, is it new method or old reported one , add reference if it is old procedure
Reply: The following statement and citation were added to the manuscript.
“The chromatographic conditions of HPLC for analysing of enrofloxacin and its metabolite ciprofloxacin were used as described by Sekkin et al. (2012).
- Provide reasons for choosing marbofloxacin as internal standard (I.S.) and why it was not measured using fluorescence detector?
Reply: Photo Diode Array Detector (DAD) was used in the analysis of marbofloxacin since marbofloxacin (I.S.) did not give sufficient absorbance at the wavelength at which enrofloxacin and its metabolite were analyzed by fluorescence detector.
- In line 396, demonstrate results of The inter- and intra-assay precisions in new table
Reply: The values for the inter- and intra-assay precisions were added to the Table 5.
- In table 5, correct Linearity (r 2 ) to be R² (coefficient of determination)
Reply: Thank you. It was corrected.
- Add missed conclusion
- Add future plan and study limitations (e.g., " Results in rabbits may not directly translate to humans").
Reply: Mentioned in the summary conclusion section
Reviewer 2 Report
Comments and Suggestions for Authors
The manuscript entitled “Food-Drug Interactions: Effect of Propolis on the Pharmacokinetics of Enrofloxacin and Its Active Metabolite Ciprofloxacin in Rabbits” is a well-designed and timely study that explores a relevant topic in the field of pharmacokinetics and natural product-drug interactions. The investigation is methodologically sound and the results are clearly presented. However, there are several issues—mostly related to clarity, language, and interpretive caution—that need to be addressed before the manuscript can be accepted for publication.
I recommend minor revision. My detailed comments are below:
1] The abstract would benefit from rephrasing for conciseness and improved clarity.
2] Several minor grammatical issues are present throughout the manuscript. A thorough proofreading or light language editing is advised to enhance readability. Examples:
Line 17: “which have many biological activities” replace with “which has...”
Line 167: “treatment with per os enrofloxacin” is awkward phrasing; consider “oral administration of enrofloxacin.”
3] Some paragraphs in the results repeat information already shown in tables. Consider condensing textual descriptions and focusing on interpretation rather than restating all numerical data.
4] Consider adding a graphical abstract that illustrates the effect of propolis on enrofloxacin pharmacokinetics to aid visual understanding.
Author Response
Author responses to the editor and reviewer’ comments
Thanks to the Editor and the reviewers for these precious comments and suggestions. These comments are all valuable and very helpful for revising and improving our paper. We have studied all the comments carefully and have made corrections which we hope to meet with approval. Revised portions are highlighted for the 1. Referee as yellow, for the 2. Referee as green, for the editor as red in the manuscript
Kindest regards
The authors of the manuscript
Referee 2
The manuscript entitled “Food-Drug Interactions: Effect of Propolis on the Pharmacokinetics of Enrofloxacin and Its Active Metabolite Ciprofloxacin in Rabbits” is a well-designed and timely study that explores a relevant topic in the field of pharmacokinetics and natural product-drug interactions. The investigation is methodologically sound and the results are clearly presented. However, there are several issues—mostly related to clarity, language, and interpretive caution—that need to be addressed before the manuscript can be accepted for publication.
I recommend minor revision. My detailed comments are below:
We would like to thank the referee for his guidance and suggestions for improving the article. The referee can see the changes highlighted in green in the manuscript.
1] The abstract would benefit from rephrasing for conciseness and improved clarity.
Reply: The abstract has been revised with the addition of some information at the request of the other referee.
2] Several minor grammatical issues are present throughout the manuscript. A thorough proofreading or light language editing is advised to enhance readability. Examples:
Reply: Changes have been made in accordance with the referee's recommendation in terms of grammar.
Line 17: “which have many biological activities” replace with “which has...”
Reply: It has been corrected
Line 167: “treatment with per os enrofloxacin” is awkward phrasing; consider “oral administration of enrofloxacin.”
Reply: Thank you, the statement was changed, as suggested.
3] Some paragraphs in the results repeat information already shown in tables. Consider condensing textual descriptions and focusing on interpretation rather than restating all numerical data.
Reply: Since there are so many tables, we tried to interpret the important data by highlighting it. Therefore, we believe that the data will be more understandable in the tables.
4] Consider adding a graphical abstract that illustrates the effect of propolis on enrofloxacin pharmacokinetics to aid visual understanding.
Reply: A graphical abstract has been prepared for the article.
Round 2
Reviewer 1 Report
Comments and Suggestions for Authors
accept